# Reduction in the Residues of Penthiopyrad in Processed Edible Vegetables by Various Soaking Treatments and Health Hazard Evaluation in China

**DOI:** 10.3390/foods12040892

**Published:** 2023-02-19

**Authors:** Jinming Chang, Li Dou, Yu Ye, Kankan Zhang

**Affiliations:** National Key Laboratory of Green Pesticide, Key Laboratory of Green Pesticide and Agricultural Bioengineering, Ministry of Education, Center for R&D of Fine Chemicals, Guizhou University, Guiyang 550025, China

**Keywords:** penthiopyrad, soaking, processing factor, edible vegetable, health risk

## Abstract

Tomato and cucumber are two vital edible vegetables that usually appear in people’s daily diet. Penthiopyrad is a new type of amide chiral fungicide, which is often used for disease control of vegetables (including tomato and cucumber) due to its wide bactericidal spectrum, low toxicity, good penetration, and strong internal absorption. Extensive application of penthiopyrad may have caused potential pollution in the ecosystem. Different processing methods can remove pesticide residues from vegetables and protect human health. In this study, the penthiopyrad removal efficiency of soaking and peeling from tomatoes and cucumbers was evaluated under different conditions. Among different soaking methods, heated water soaking and water soaking with additives (NaCl, acetic acid, and surfactant) presented a more effective reduction ability than other treatments. Due to the specific physicochemical properties of tomatoes and cucumbers, the ultrasound enhances the removal rate of soaking for tomato samples and inhibits it for cucumber samples. Peeling can remove approximately 90% of penthiopyrad from contaminated tomato and cucumber samples. Enantioselectivity was found only during tomato sauce storage, which may be related to the complex microbial community. Health risk assessment data suggests that tomatoes and cucumbers are safer for consumers after soaking and peeling. The results may provide consumers with some useful information to choose better household processing methods to remove penthiopyrad residues from tomatoes, cucumbers, and other edible vegetables.

## 1. Introduction

Edible vegetables are a type of vegetable that can be consumed raw or cooked. Most edible vegetables contain a lot of nutrient substances and vitamins and eating nutrient-rich edible vegetables can successfully decrease the risk of some illnesses [1]. Tomato and cucumber are two vital edible vegetables with unusually tasty and nutritious properties containing four major carotenoids and three highly potent antioxidants [2]. Eating raw or cooked tomatoes and cucumber may be very efficient in prohibiting prostate and pancreatic cancer [3]. Moreover, these two edible vegetables were some of the most economically valuable crops and have been cultivated on a large scale worldwide [4]. Nevertheless, similar to other edible vegetables, the growth of tomato and cucumber may be easily hindered by some bacterial or fungal diseases (such as botrytis, powdery mildew, rust, and so on) which could greatly decrease the yield and further alter the later processing products [5]. The application of agrochemicals is a critical approach to effectively control diseases in the modern cultivation and production of edible vegetables [6].

Penthiopyrad, a chiral succinate dehydrogenase inhibitor (SDHI) fungicide, has a broad-spectrum anti-bactericidal activity for tomatoes, cucumbers, and other edible vegetables. It can effectively control powdery mildew, fruit tree black spot, rust, black star fungus, and gray mold [7,8,9,10]. Extensive use of penthiopyrad may have caused the accumulation and residues in raw and processed tomato and cucumber samples. In our previous study, the distribution and dissipation of penthiopyrad have been demonstrated in the field (raw) tomato and cucumber samples [11]. However, data on the residues of processed penthiopyrad-contaminated tomato and cucumber samples and the removal efficiency of the processing methods are still lacking.

In order to ensure food safety and protect human health, it is important to eliminate agrochemical residues in vegetables [12]. Some household processing methods can effectively remove agrochemical residues in raw vegetables [13]. Among them, washing is the simplest and most easy to operate processing approach, constantly the first step in the household processing procedures, and may be the only step for raw edible vegetables [14,15,16]. However, rinsing with tap water alone was not effective in removing agrochemical residues, as rinsing only reduced the levels of agrochemicals that loosely attached to vegetable surfaces [17]. Soaking in tap water or different non-toxic solutions (organic, inorganic, or surfactant solvents) could significantly improve the removal efficiency of agrochemical residues [18,19,20]. Furthermore, some physical-assisted methods including peeling, ultrasound, and heating can also enhance the reduction ability of agrochemicals from vegetables after soaking [21,22]. Therefore, a systemic study of the effects of different soaking methods for removing agrochemical residues from edible vegetables could provide data and technical support for food safety and human health.

Although the dissipation residue and dietary risk results of penthiopyrad in field tomato and cucumber samples showed that the recommended application approach did not threaten vegetable cultivation and human health [11], the potential pollution risk of penthiopyrad in raw and processed edible vegetable samples will occur due to the accumulation after long-term and wide application. Therefore, in our current study, the removal efficiency of penthiopyrad residues from tomato and cucumber samples by regular and ultrasound-assisted soaking has been evaluated. The removal of penthiopyrad was also investigated by regular soaking in different solutions (heated aqueous, inorganic salt, acid, surfactant, and organic solutions). Meanwhile, among the above trials, the distribution and health hazard risk of penthiopyrad in different parts of two edible vegetables (pulp, peel, and whole fruit) and the persistence in tomato and cucumber productions (sauce or juice) were illustrated.

## 2. Materials and Methods

### 2.1. Chemicals and Reagents

Racemic penthiopyrad (purity, 99.5%), manufactured by Dr. Ehrenstorfer GmbH (Augsburg, Germany), was purchased from J&K Scientific Ltd. (Beijing, China). Chromatographic grade acetonitrile (ACN, 99.9%), methanol (MeOH, 99.9%), and formic acid (FA, 8.0%) were obtained from Thermo Fisher Scientific (Waltham, MA, USA). Analytical grade sodium chloride (NaCl, 99.5%), anhydrous sodium sulfate (Na_2_SO_4_, 99.0%), ethanol (EA, 99.7%), linear alkylbenzene sulfonates (LAS, 99.0%), and acetic acid (AA, 99.5%) were provided by Tianjin Zhiyuan Reagent Co., Ltd. (Tianjin, China). Primary secondary amine (PSA, 99.5%) was bought from Agela Technologies (Tianjin, China), and a 0.22 μm nylon syringe filter was obtained from PeakSharp Technologies (Beijing, China).

### 2.2. Instrumentation

Chromatographic separation and quantification of penthiopyrad enantiomers in different matrices were performed on a liquid chromatography with tandem mass spectrometry (LC-MS/MS) [11]. The conditions of the LC system (Shimadzu Corporation, Kyoto, Japan) included: column, Superchiral S-OD chiral column (150 × 4.6 mm i.d.; particle size, 5 µm; Shanghai Chiralway Biotech Co., Ltd., Shanghai, China); mobile phase, 0.1% FA in ACN/0.1% FA in aqueous solution (50/50, *v*/*v*); flow rate, 1.0 mL/min; injection volume, 3 μL; column temperature, 30 °C; retention time, 11 min. The conditions of the MS system (Applied Biosystems, Foster City, CA, USA) were: gas source, nitrogen (99.999%); ion source temperature, 600 °C; ion source gas 1 and 2 pressure, 448 kPa; ion spray voltage, 5.5 kV; detection mode, positive electron spray ionization, and multiple reaction monitoring; *m*/*z* (parent ion→daughter ion), 360.3→256.0 for confirmation and 360.3→276.0 for quantification; declustering potential, 103.5 V; entrance potential, 10.0 V; collision energy and collision cell exit potential, 28.7 V and 15.7 V for daughter ion 256.0 and 18.4 V and 25.9 V for daughter ion 276.0, respectively.

### 2.3. Pretreatment

Raw and processed tomato and cucumber samples were thoroughly crushed and homogenized for subsequent analysis. The pretreatment method was a modified QuEChERS (quick, easy, cheap, effective, rugged, and safe) approach which has been reported in our previous research [11]. Ten grams of tomato (or cucumber) samples were weighed in a centrifuge tube and then 20 mL of ACN, 2 g of NaCl, and 4 g of anhydrous Na_2_SO_4_ were added. After vortex-mixing for 5 min and centrifuging for 5 min at 1677× *g*, 1 mL of supernatant was transferred to a centrifuge tube containing 50 mg of PSA. The mixture was vortexed for 0.5 min, centrifuged for 3 min at 6708× *g,* and filtered through a nylon filter syringe before instrument analysis.

### 2.4. Processing Experiments

Raw (no penthiopyrad applied) tomato and cucumber samples were purchased from the local market and used in the following processing trials. Before processing, penthiopyrad-contaminated tomato and cucumber samples should be preferentially obtained by dipping raw edible vegetables in penthiopyrad solutions [23]. Dipping experiments were performed in 10-L containers with 5 L of penthiopyrad solution (0.5mg/L), allowing the raw samples to be fully immersed for 60 min. The penthiopyrad-contaminated samples were then air-dried on absorbent paper. A set of control tomato and cucumber samples were analyzed while evaluating the removal efficiency of penthiopyrad by different processing methods. There are three replicates of each treatment.

#### 2.4.1. Soaking and Peeling

Different soaking methods have been used, including water soaking, heated water soaking, ultrasound-assisted water soaking, and water soaking with additives. In addition to the whole fruit samples of processed tomatoes and cucumbers, peel and pulp samples of two edible vegetables were analyzed after regular peeling in all soaking treatments. The parameters of each soaking method were listed in Appendix A and the specific operations were listed as follows:

(1) Water soaking and peeling: Blank and penthiopyrad-contaminated tomato or cucumber samples were soaked in a container with water at room temperature. The soaking durations were 5, 10, 15, and 20 min. The processed whole fruit, peel, and pulp (after peeling) samples were gathered from each treatment.

(2) Heated water soaking and peeling: Blank and penthiopyrad-contaminated tomato or cucumber samples were immersed in a container with heated water (maintained at 45 °C). The soaking durations were 5, 10, 15, and 20 min. The processed whole fruit, peel, and pulp (after peeling) samples were obtained from each treatment.

(3) Ultrasound-assisted water soaking and peeling: Blank and penthiopyrad-contaminated tomato or cucumber samples were soaked in ultrasound equipment with water at room temperature. The ultrasound frequency was set to 40 kHz. The soaking durations were 4, 6, 8, and 10 min (avoid raising the water temperature due to the long operation of the instrument). The processed whole fruit, peel, and pulp (after peeling) samples were collected from each treatment.

(4) Water soaking with additives and peeling: Blank and penthiopyrad-contaminated tomato or cucumber samples were immersed at room temperature in a container containing aqueous solutions of different additives. The additives include NaCl, AA, LAS, and EA, with an additive content of 0.2%, 0.5%, and 1%. The soaking duration was 15 min. The processed whole fruit, peel, and pulp (after peeling) samples were gathered from each treatment.

#### 2.4.2. Saucing (Tomato) and Juicing (Cucumber)

Saucing and juicing are two deep processing methods for edible vegetables. In this study, after soaking in water at room temperature and peeling, saucing was selected to produce tomato sauce and juicing was chosen to produce cucumber juice. The persistence of penthiopyrad in tomato sauce and cucumber juice was investigated by storing at room temperature and 4 °C. After processing, samples were collected at 0, 1, 2, 3, 5, 7, 9, and 12 d. The detailed procedure is shown in Appendix A.

### 2.5. Data Analysis

The residue concentrations of penthiopyrad enantiomers in the tomato and cucumber were calculated by comparing the peak area of the treated samples to the matrix-matched standard solution. The enantiomeric fraction (EF) value was used to evaluate the enantioselectivity of penthiopyrad in processed edible vegetable products and calculated as the ratio of the level of *R*-(−)-enantiomer (penthiopyrad enantiomer with an R spatial configuration) to the level of racemic penthiopyrad (*Rac*-penthiopyrad) [11]. The removal rate (RR), one value to evaluate the effect of the processing method, is determined by the equation: RR (%) = (level of penthiopyrad in raw edible vegetable—level in processed edible vegetable)/level in processed edible vegetable × 100%. The processing factor (PF), another parameter to illustrate the removal efficiency of the process approach, is defined as the ratio of the residue found in the processed commodity to that in the raw agricultural commodity before processing. PF < 1 and PF > 1 indicated a decrease and an increase in agrochemical residue levels during processing, respectively [24,25]. The hazard quotient (HQ) value is used to evaluate the health risk of penthiopyrad in processed edible vegetable samples [26]. Two parameters, NESTI (national estimated short-term intake) and ARfD (acute reference dose suggested for daily exposure from short-term intake) were applied to calculated HQ values (NESTI/ARfD × 100%) of penthiopyrad in processed tomato and cucumber samples for Chinese consumers. NESTI is related to the food consumption intake obtained from the survey report of Chinese residents’ dietary (FI), residue of *Rac*-penthiopyrad in samples (C), and the body weight of consumer (b.w.), which is determined from the equation NESTI = FI × C/b.w. [27,28]. The ARfD of penthiopyrad is 1 mg/kg b.w. [29]. All data are expressed as mean ± standard deviation (SD). *p*-values of less than 0.05 were deemed statistically significant. SPSS version 18.0 (SPSS Inc. Chicago, IL, USA) was used for statistical analysis [30].

## 3. Results

### 3.1. Effects of Different Soaking Treatments to Remove Penthiopyrad from Two Edible Vegetables

The residues of penthiopyrad enantiomers and racemate in processed tomato samples after different soaking methods are shown in Appendix A. The initial levels of *R*-, *S*- and *Rac*-penthiopyrad in different parts of tomato were 161–323 µg/kg (pulp), 2261–4555 µg/kg (peel), and 753–1514 µg/kg (whole fruit), which showed that peeling was still one of the most effective processing approaches for removing penthiopyrad residues from tomato samples. After soaking in water (treatment A), the RR values (Figure 1A) for penthiopyrad increased from 86.2% to 97.7% in pulp, from 8.6% to 35.0% in peel and from 8.9% to 32.9% in whole fruit with the extension of soaking duration. When the soaking solution was heated water (treatment B) or the soaking was assisted by ultrasound (treatment C), the RR values (Figure 1B,C) were 1.6–3.8 times more than those at the corresponding soaking duration in treatment A. Four additives (NaCl, AA, LAS, and EA) with different contents were applied and the levels of penthiopyrad enantiomers and racemate in raw and processed tomato samples are listed in Appendix A, respectively. The RR data in Figure 1D–G, with a 1% of additive, offered the best removal efficiency for all four chemicals. For example, in whole fruit samples, the RR values for *Rac*-penthiopyrad were 25.6% in treatment A, and 74.2% (0.2%), 86.4% (0.5%), and 89.2% (1%) in treatment D. The removal efficiency of water soaking with 1% of additives reached more than 40% in treatment G and >70% in the remaining three treatments. In addition, PF values were calculated for different soaking treatments and are shown in Table 1 and Table 2. All PF values were less than 1.0 and ranged from 0.91 to 0.02.

The residues of penthiopyrad enantiomers and racemate in raw and soaking-processed cucumber samples are presented in Appendix A. The initial concentrations of *R*-, *S*-, and *Rac*-penthiopyrad in raw cucumber samples (whole fruit) were in 2012, 2008, and 4021 µg/kg, respectively. After peeling, the initial levels of penthiopyrad were 304–608 µg/kg in pulp samples and 5875–10540 µg/kg in peel samples. In addition to the tomato samples, peeling also provided a large reduction in penthiopyrad residues in the cucumber samples. In Figure 2A,C, the RR values are 9.8–96.2% for treatment A, 69.2–97.9% for treatment B, and 16.2–94.3% for treatment C. When soaking in water with additives, the concentrations of penthiopyrad in cucumber samples also decreased (Appendix A. The addition of NaCl, AA, and LAS made the reduction of penthiopyrad residues more efficient after soaking in all three tests with an increase of RR value (Figure 2D,F). An opposite result was found in treatment G that after adding EA the reduction of penthiopyrad enantiomers was inhibited in processed cucumber samples (RR values decreased 4.8–37.5%, Figure 2G).

In Table 3 and Table 4, PF values of water soaking with 1% NaCl, 1% AA, 1% LAS, and 1% EA are 0.03, 0.01–0.02, 0.01, and 0.46–0.47 for whole fruit cucumber samples. Compared to the data in water soaking without additive (PF values of 0.39–0.40), NaCl, AA, and LAS significantly improved the relief ability of water soaking and EA inhibited that ability. In summary, peeling and soaking can remove penthiopyrad residues from tomatoes and cucumbers, and temperature and the addition of chemicals can significantly affect the removal efficiency of soaking.

### 3.2. Effects of Saucing or Juicing to Remove Penthiopyrad from Two Edible Vegetables and Their Storage Stability

In the peeling and soaking trial, peeling removed almost 90 percent of penthiopyrad residues from contaminated tomato and cucumber samples. The initial concentrations of *R*-, *S*- and *Rac*-penthiopyrad were 162, 161, and 323 µg/kg in tomato pulp samples, respectively, and were 305, 304, and 608 µg/kg in cucumber pulp samples, respectively. The levels of *R*-, *S*- and *Rac*-penthiopyrad reduced to 161, 145, and 306 µg/kg, respectively, in tomato sauce samples after saucing and reduced to 63, 64, and 128 µg/kg, respectively, in cucumber juice samples after juicing (Appendix A. In Figure 1H, the concentrations of penthiopyrad decreased by more than 36% in tomato sauce samples stored for 12 d at 4 °C and room temperature. However, during the 12-day storage experiment, the RR values ranged from 0.5% to 14.8% at 4 °C and from 2.5% to 22.9% at room temperature in stored cucumber juice samples (Figure 2H). Additionally, PF values (Figure 3) of *Rac*-penthiopyrad decreased from 0.92 to 0.62 (4 °C) and from 0.82 to 0.59 (room temperature) in tomato sauce samples during storage. Those in cucumber juice samples decreased from 0.99 to 0.86 (4 °C) and from 0.97 to 0.78 (room temperature).

### 3.3. Potential Enantioselectivity and Health Risk of Penthiopyrad in Tomato and Cucumber after Processing

As a chiral fungicide, penthiopyrad may experience enantioselective reduction during the processing. However, except in the tomato sauce storage trial, no enantioselectivity of penthiopyrad was found in other processing treatments because most EF values were close to 0.5 (Figure 4). The HQ values of racemic penthiopyrad in processed tomato and cucumber samples after different soaking approaches for different gender and age groups of Chinese consumers are shown in Appendix A. For tomato and cucumber pulp samples, the HQ values were <0.2%. In whole fruit samples, the HQ values of *Rac*-penthiopyrad were <0.5% (tomato) and <1.3% (cucumber). In addition, the higher HQ values were found in tomato (1.4%) and cucumber (3.7%) peel samples The above health risk results demonstrate that water soaking with AA and LAS has the highest removal efficiency for penthiopyrad residues in tomato and cucumber samples.

## 4. Discussion

In the current study, the removal efficiency of different soaking (with or without peeling) methods was investigated for penthiopyrad from two edible vegetables. The reduction data illustrates that the soaking duration can improve the removal efficiency of the soaking method for penthiopyrad from tomato samples. Various soaking approaches have been proposed for the target compound with various removal efficiencies. Water (unheated) soaking could remove a part of penthiopyrad residues from tomatoes and cucumbers and the reduction amount increased in heated water soaking treatment, which demonstrated that temperature could affect the removal of soaking for penthiopyrad. Temperature and ultrasound were positively correlated to the removal efficiency of soaking for penthiopyrad in tomatoes. Kaushik et al. [31] reported that washing tomato samples in warm water could increase the relief of tetranilipole by approximately 1.3 times. Heshmati and Nazemi [32] studied the effect of ultrasound-assisted washing to remove the dichlorvos residues from tomato samples and found an increase in removal under ultrasonic conditions. However, the residues of penthiopyrad in tomato pulp samples have not reduced more after treatments B and C, possibly due to the physicochemical properties of penthiopyrad [20]. The exact reasons are unclear and need to be revealed in our future studies. Adding some chemicals to water could change the removal efficiency of soaking for pesticides from vegetable samples [33]. The residues of penthiopyrad enantiomers decreased after soaking in water with NaCl (treatment D), AA (treatment E), LAS (treatment F), and EA (treatment G), which meant these chemicals could remove penthiopyrad from tomato by soaking. Previous papers illustrated that after adding some additives (including organic acid, chlorine-based, and detergent) washing could remove more pesticide residues from tomato [34] and citrus samples [35]. In the present research, we found that water soaking with organic acid (AA), NaCl, and surfactant (LAS) showed excellent relief ability for penthiopyrad enantiomers from tomato samples. For EA, the removal efficiency did not improve a lot, which might be related to the polarity of fungicides and chemicals [3]. All the PF values < 1 indicated that the soaking was able to remove penthiopyrad residues from different parts of the tomato samples regardless of the experimental conditions, and statistical analysis of the data revealed that treatment A-G was able to reduce the concentration of penthiopyrad from the contaminated tomato sample, demonstrating a significant difference in the treatment of certain soaking.

For cucumbers, the removal efficiency is significantly improved when the temperature is increased. However, ultrasound has little effect on increasing penthiopyrad reduction and can even inhibit the removal efficiency. The PF values provide further evidence that temperature significantly affects the removal rate of soaking for penthiopyrad from the cucumber samples and the ultrasound-assisted soaking might be an influence, but not an important one. The possible reason is the special physicochemical properties of cucumber. The cucumber surface has a lot of uneven places, which favors the accumulation of pesticides. While ultrasound accelerates the reduction of pesticides, it also accelerates the adsorption and accumulation of pesticides on the uneven surface of cucumbers. When the soaking duration was overly prolonged, the accumulation rate was greater than the reduction rate, and finally decreased the removal efficiency of the water soaking method [20,36]. The addition of chemicals reduced the removal of soaking for penthiopyrad from cucumber samples because of the physicochemical properties of penthiopyrad and EA [22,37]. However, the addition of NaCl, AA, and LAS was found to improve the removal rate for penthiopyrad from cucumber samples in the waster soaking with additive trials. As the additive content increases, the removal efficiency improves.

Peeling was able to remove most of the fungicide residue from the contaminated tomato and cucumber samples due to the large reduction in the penthiopyrad levels in the pulp samples. After peeling, tomato sauce and cucumber juice are produced from the pulp of the respective edible vegetables. The peeling results indicated that most of the penthiopyrad residues in tomato and cucumber samples were found in the solid part after squeezing [12]. Due to the specific biochemical and microbiological characteristics of the two edible vegetable products, the reduction of penthiopyrad was different in tomato sauce and cucumber juice samples [33]. Tomato sauce contains many acetic compounds, minerals, and pectin, which can remove pesticides in several ways [38]. Unlike cucumber juice, which contains only vitamins and dietary fiber, there is little relief from pesticides. The PF data also indicates that storage may remove a certain amount of penthiopyrad, and the effect is more significant in the tomato sauce sample [1].

An enantioselective dissipation of penthiopyrad has been illustrated in tomato and cucumber samples cultivated under field and greenhouse conditions in our previous study [11]. During the storage of tomato sauce, the reduction of penthiopyrad was enantioselective (*R*-enantiomer reduced preferentially) in tomato sauce samples due to EF values > 0.5. The enantioselectivity was perhaps related to the complex microbial community in tomato sauce [28]. Because tomatoes and cucumbers are often washed and then eaten by humans, consumers should be concerned about the health risks of penthiopyrad. For tomato and cucumber pulp samples, the health risk could be negligible for consumers, meaning that after soaking and peeling, the two edible vegetables are safe for human consumption. For whole fruit samples, a consumer could eat two edible vegetables that have only been soaked. High HQ values in peel samples indicated that peeling was an effective processing method to remove penthiopyrad residues from tomato and cucumber samples. The results indicate that the above processing methods to remove penthiopyrad residues from vegetables are crucial to protect human health, strengthen the awareness of food hygiene and safety of consumers, and prevent the occurrence of potential food safety cases.

## 5. Conclusions

In summary, the effects of different processing methods (soaking, peeling, juicing, and saucing) on penthiopyrad residues were investigated in two edible vegetables (tomato and cucumber). The results showed that water soaking at room temperature could remove a part of penthiopyrad residues from tomatoes (RR < 33%) and cucumbers (RR < 80%) and the reduction amount for tomatoes (RR > 60%) and cucumbers (RR > 90%) increased in heated water soaking treatment. The ultrasound-assisted soaking method showed different efficiency for reducing penthiopyrad concentrations in tomato (improvement, RR > 70%) and cucumber (inhibition, RR < 62%) samples. PF values (decreased from 0.39 to 0.01) indicated that the addition of NaCl, AA, and LAS improved the removal rate for penthiopyrad from tomato and cucumber samples in the soaking trials. However, after the addition of EA, the removal efficiency of the soaking of both edible vegetables decreased (PF increased from 0.39 to 0.47). After 12 days of storage, the levels of penthiopyrad were reduced by >36% in the tomato sauce sample and 13.3% to 22.9% in the cucumber juice sample. The reduction of penthiopyrad after most processing treatments was not enantioselective (EF ≈ 0.5) and a preferential removal of *R*-penthiopyrad was found during the storage process of tomato sauce. In addition, HQ data show that consumers can eat the soaked tomatoes and cucumbers and that it is safer to peel both edible vegetables. The results will provide some information and data on the removal of penthiopyrad from other edible vegetables, and help consumers choose the most effective and simple processing method for reducing pesticide residues in tomatoes and cucumbers.

## Figures and Tables

**Figure 1 foods-12-00892-f001:**
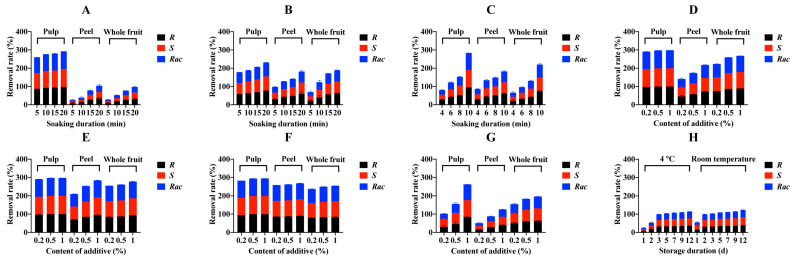
Rates of different soaking methods (**A**): water soaking and peeling; (**B**): heated water soaking and peeling; (**C**): ultrasound-assisted soaking and peeling; (**D**): water soaking with NaCl and peeling; (**E**): water soaking with AA and peeling; (**F**): water soaking with LAS and peeling; (**G**): water soaking with EA and peeling) and storing process at 4 °C and room temperature for different durations after saucing, (**H**) to remove penthiopyrad enantiomers from tomato samples.

**Figure 2 foods-12-00892-f002:**
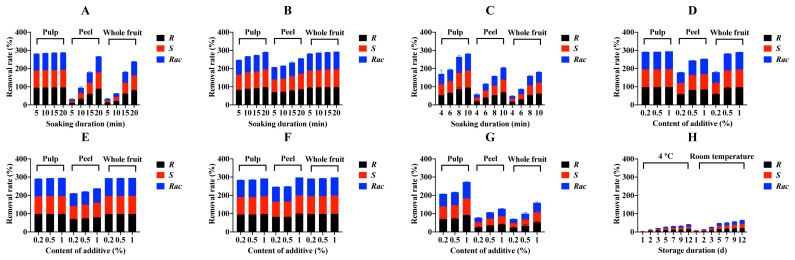
Rates of different soaking methods ((**A**): water soaking and peeling; (**B**): heated water soaking and peeling; (**C**): ultrasound-assisted soaking and peeling; (**D**): water soaking with NaCl and peeling; (**E**): water soaking with AA and peeling; (**F**): water soaking with LAS and peeling; (**G**): water soaking with EA and peeling) and storing process at 4 °C and room temperature for different durations after juicing (**H**) to remove penthiopyrad enantiomers from cucumber samples.

**Figure 3 foods-12-00892-f003:**
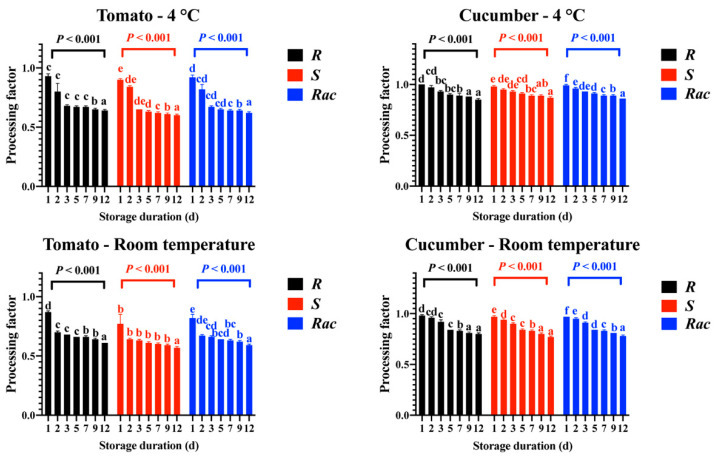
PF values of *R*-, *S*-, and *Rac*-penthiopyrad in tomato sauce and cucumber juice samples after storing at different durations under 4 °C and room temperature. Each lowercase letter indicates significant differences between the six sampling intervals by Duncan’s multiple range test (*p* = 0.01).

**Figure 4 foods-12-00892-f004:**
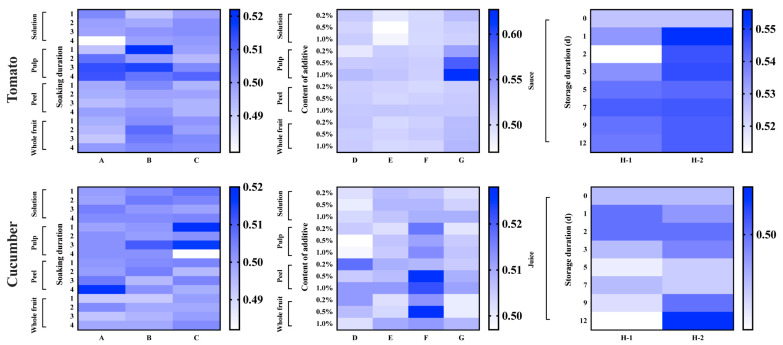
Heatmap of EF values of penthiopyrad in processed tomato and cucumber samples by different soaking methods (A: water soaking and peeling (1, 2, 3, 4 represented 5, 10, 15, 20 min); B: heated water soaking and peeling (1, 2, 3, 4 represented 5, 10, 15, 20 min); C: ultrasound-assisted soaking and peeling (1, 2, 3, 4 represented 4, 6, 8, 10 min); D: water soaking with NaCl and peeling; E: water soaking with AA and peeling; F: water soaking with LAS and peeling; G: water soaking with EA and peeling) and saucing or juicing in different durations at 4 °C (H-1) and room temperature (H-2).

**Table 1 foods-12-00892-t001:** Processing factor (PF) values of penthiopyrad enantiomers in tomato samples after different soaking methods.

Treatment	Duration (min)	Additive Content	PF (Average ± SD, *n* = 3)											
			Pulp				Peel				Whole Fruit			
			*R*	*p*	*S*	*p*	*R*	*p*	*S*	*p*	*R*	*p*	*S*	*p*
A	5	/	0.13 ± 0 d	<0.001	0.14 ± 0.01 d	<0.001	0.91 ± 0.03 b	<0.001	0.91 ± 0.03 c	<0.001	0.90 ± 0.02 c	<0.001	0.91 ± 0.03 d	<0.001
	10	/	0.08 ± 0 c		0.08 ± 0 c		0.87 ± 0.03 b		0.87 ± 0.05 bc		0.82 ± 0.02 c		0.83 ± 0.03 c	
	15	/	0.07 ± 0.01 b		0.06 ± 0.01 b		0.74 ± 0.03 a		0.75 ± 0.02 ab		0.73 ± 0.01 b		0.75 ± 0.01 b	
	20	/	0.02 ± 0 a		0.02 ± 0 a		0.66 ± 0.07 a		0.65 ± 0.07 a		0.68 ± 0.05 a		0.67 ± 0.02 a	
B	5	/	0.42 ± 0.01 d	<0.001	0.39 ± 0.01 c	<0.001	0.68 ± 0.03 c	<0.001	0.67 ± 0.03 c	<0.001	0.78 ± 0.05 c	<0.001	0.76 ± 0.07 c	<0.001
	10	/	0.37 ± 0.01 c		0.38 ± 0.02 b		0.57 ± 0.01 b		0.57 ± 0 b		0.60 ± 0.10 c		0.58 ± 0.11 bc	
	15	/	0.32 ± 0.01 b		0.30 ± 0 a		0.53 ± 0.02 ab		0.53 ± 0.02 ab		0.43 ± 0.01 b		0.42 ± 0 b	
	20	/	0.23 ± 0.03 a		0.23 ± 0.02 a		0.40 ± 0.07 a		0.39 ± 0.08 a		0.37 ± 0.02 a		0.37 ± 0.02 a	
C	4	/	0.73 ± 0.01 d	<0.001	0.73 ± 0.01 d	<0.001	0.71 ± 0.03 c	<0.001	0.71 ± 0.04 c	<0.001	0.79 ± 0.05 c	<0.001	0.78 ± 0.06 c	<0.001
	6	/	0.58 ± 0.02 c		0.60 ± 0.05 c		0.55 ± 0.03 b		0.55 ± 0.03 b		0.68 ± 0.02 b		0.68 ± 0.02 b	
	8	/	0.49 ± 0.01 b		0.49 ± 0.02 b		0.50 ± 0.03 b		0.51 ± 0.03 b		0.57 ± 0.04 ab		0.56 ± 0.05 ab	
	10	/	0.06 ± 0.01 a		0.05 ± 0.01 a		0.39 ± 0.03 a		0.39 ± 0.03 a		0.27 ± 0.05 a		0.26 ± 0.05 a	
D	15	0.2%	0.03 ± 0 b	<0.001	0.03 ± 0.01 a	0.026	0.54 ± 0.06 c	<0.001	0.52 ± 0.05 c	<0.001	0.27 ± 0.02 b	<0.001	0.25 ± 0.02 b	<0.001
	15	0.5%	0.02 ± 0 b		0.02 ± 0 a		0.43 ± 0.02 b		0.41 ± 0.02 b		0.14 ± 0 b		0.13 ± 0 b	
	15	1%	0.02 ± 0 a		0.02 ± 0 a		0.30 ± 0.01 a		0.28 ± 0.02 a		0.11 ± 0 a		0.10 ± 0 a	
E	15	0.2%	0.03 ± 0 b	0.001	0.03 ± 0 b	0.002	0.31 ± 0.01 c	<0.001	0.30 ± 0.02 c	<0.001	0.15 ± 0 b	<0.001	0.15 ± 0.01 b	<0.001
	15	0.5%	0.03 ± 0 a		0.03 ± 0 ab		0.16 ± 0 b		0.16 ± 0 b		0.13 ± 0.01 a		0.13 ± 0.01 a	
	15	1%	0.02 ± 0 a		0.02 ± 0 a		0.06 ± 0.0 a1		0.05 ± 0.0 a1		0.08 ± 0.01 a		0.08 ± 0.01 a	
F	15	0.2%	0.06 ± 0 b	0.002	0.06 ± 0 b	0.003	0.14 ± 0 a	0.052	0.14 ± 0 b	0.022	0.21 ± 0.01 b	<0.001	0.21 ± 0.02 b	0.002
	15	0.5%	0.05 ± 0 b		0.05 ± 0 b		0.13 ± 0.01 a		0.13 ± 0 ab		0.17 ± 0 b		0.16 ± 0.01 b	
	15	1%	0.05 ± 0 a		0.05 ± 0 a		0.12 ± 0.01 a		0.11 ± 0.02 a		0.15 ± 0 a		0.15 ± 0 a	
G	15	0.2%	0.75 ± 0.02 c	<0.001	0.64 ± 0.03 c	<0.001	0.85 ± 0.02 c	<0.001	0.81 ± 0.02 c	<0.001	0.51 ± 0.05 b	0.003	0.46 ± 0.04 b	0.002
	15	0.5%	0.59 ± 0.05 b		0.42 ± 0.07 b		0.73 ± 0.02 b		0.69 ± 0.02 b		0.41 ± 0.01 b		0.37 ± 0.01 b	
	15	1%	0.17 ± 0 a		0.10 ± 0.01 a		0.60 ± 0.02 a		0.56 ± 0.04 a		0.37 ± 0 a		0.33 ± 0 a	

A: Water soaking and peeling. B: Heated water soaking and peeling. C: Ultrasound-assisted soaking and peeling. D: Water soaking with NaCl and peeling. E: Water soaking with AA and peeling. F: Water soaking with LAS and peeling. G: Water soaking with EA and peeling. *p* represented confidence within the 99% confidence level of concentrations of penthiopyrad enantiomers, different lowercase letters indicate statistical significance between different treatments for tomato by Duncan’s multiple range test (*p* = 0.01).

**Table 2 foods-12-00892-t002:** PF values of racemic penthiopyrad in tomato samples after different soaking methods.

Treatment	Duration (min)	Additive Content	PF (Average ± SD, *n* = 3)					
			Tomato					
			Pulp	*p*	Peel	*p*	Whole Fruit	*p*
A	5	/	0.14 ± 0.01 d	<0.001	0.91 ± 0.03 b	<0.001	0.91 ± 0.02 d	<0.001
	10	/	0.08 ± 0 c		0.87 ± 0.04 b		0.83 ± 0.02 c	
	15	/	0.07 ± 0.01 b		0.74 ± 0.03 a		0.74 ± 0.01 b	
	20	/	0.02 ± 0 a		0.65 ± 0.07 a		0.68 ± 0.03 a	
B	5	/	0.41 ± 0.01 c	<0.001	0.68 ± 0.03 c	<0.001	0.77 ± 0.06 c	<0.001
	10	/	0.37 ± 0.02 b		0.57 ± 0.01 b		0.59 ± 0.11 bc	
	15	/	0.31 ± 0.01 a		0.53 ± 0.02 ab		0.43 ± 0 b	
	20	/	0.23 ± 0.02 a		0.39 ± 0.07 a		0.37 ± 0.02 a	
C	4	/	0.73 ± 0.01 d	<0.001	0.71 ± 0.03 c	<0.001	0.78 ± 0.06 c	<0.001
	6	/	0.59 ± 0.03 c		0.55 ± 0.03 b		0.68 ± 0.02 b	
	8	/	0.49 ± 0.02 b		0.50 ± 0.03 b		0.56 ± 0.04 ab	
	10	/	0.06 ± 0.01 a		0.39 ± 0.03 a		0.27 ± 0.05 a	
D	15	0.2%	0.03 ± 0.01 b	0.006	0.53 ± 0.05 c	<0.001	0.26 ± 0.02 b	<0.001
	15	0.5%	0.02 ± 0 ab		0.42 ± 0.02 b		0.14 ± 0 b	
	15	1%	0.02 ± 0 a		0.29 ± 0.01 a		0.11 ± 0 a	
E	15	0.2%	0.03 ± 0 b	0.001	0.30 ± 0.01 c	0.002	0.15 ± 0.01 b	<0.001
	15	0.5%	0.03 ± 0 a		0.16 ± 0 b		0.13 ± 0 a	
	15	1%	0.02 ± 0 a		0.06 ± 0.01 a		0.08 ± 0.01 a	
F	15	0.2%	0.06 ± 0 b	0.001	0.14 ± 0 a	0.033	0.21 ± 0.02 b	0.001
	15	0.5%	0.05 ± 0 b		0.13 ± 0.01 a		0.17 ± 0 b	
	15	1%	0.05 ± 0 a		0.11 ± 0.02 a		0.15 ± 0 a	
G	15	0.2%	0.69 ± 0.02 c	<0.001	0.83 ± 0.02 c	<0.001	0.49 ± 0.05 b	<0.001
	15	0.5%	0.50 ± 0.06 b		0.71 ± 0.02 b		0.39 ± 0.01 b	
	15	1%	0.13 ± 0 a		0.58 ± 0.02 a		0.35 ± 0 a	

A: Water soaking and peeling. B: Heated water soaking and peeling. C: Ultrasound-assisted soaking and peeling. D: Water soaking with NaCl and peeling. E: Water soaking with AA and peeling. F: Water soaking with LAS and peeling. G: Water soaking with EA and peeling. *p* represented confidence within the 99% confidence level of concentrations of penthiopyrad enantiomers, different lowercase letters indicate statistical significance between different treatments for tomato by Duncan’s multiple range test (*p* = 0.01).

**Table 3 foods-12-00892-t003:** PF values of penthiopyrad enantiomers in cucumber samples after different soaking methods.

Treatment	Duration (min)	Additive Content	PF (Average ± SD, *n* = 3)											
			Pulp				Peel				Whole Fruit			
			*R*	*p*	*S*	*p*	*R*	*p*	*S*	*p*	*R*	*p*	*S*	*p*
A	5	/	0.06 ± 0 c	<0.001	0.06 ± 0 c	<0.001	0.89 ± 0.03 d	<0.001	0.89 ± 0.04 d	<0.001	0.87 ± 0.02 d	<0.001	0.90 ± 0.02 c	<0.001
	10	/	0.05 ± 0 c		0.05 ± 0 c		0.68 ± 0.06 c		0.68 ± 0.05 c		0.79 ± 0.01 c		0.79 ± 0.02 d	
	15	/	0.04 ± 0 b		0.04 ± 0 b		0.40 ± 0.03 b		0.40 ± 0.04 b		0.39 ± 0.03 b		0.40 ± 0.03 b	
	20	/	0.04 ± 0 a		0.04 ± 0 a		0.11 ± 0 a		0.11 ± 0 a		0.20 ± 0.01 a		0.21 ± 0.02 a	
B	5	/	0.17 ± 0 d	<0.001	0.17 ± 0.01 d	<0.001	0.31 ± 0 c	<0.001	0.31 ± 0 c	<0.001	0.06 ± 0.01 c	<0.001	0.06 ± 0.01 c	<0.001
	10	/	0.11 ± 0 c		0.11 ± 0 c		0.28 ± 0.01 b		0.28 ± 0.05 b		0.04 ± 0 bc		0.04 ± 0 bc	
	15	/	0.09 ± 0.01 b		0.09 ± 0 b		0.22 ± 0.03 a		0.23 ± 0.01 ab		0.03 ± 0 b		0.03 ± 0 b	
	20	/	0.03 ± 0 a		0.03 ± 0 a		0.15 ± 0 a		0.18 ± 0.01 a		0.02 ± 0 a		0.02 ± 0 a	
C	4	/	0.45 ± 0.23 b	0.014	0.41 ± 0.19 c	0.007	0.81 ± 0.04 d	<0.001	0.80 ± 0.04 d	<0.001	0.83 ± 0.03 d	<0.001	0.84 ± 0.03 d	<0.001
	6	/	0.35 ± 0.03 ab		0.35 ± 0.04 bc		0.60 ± 0.01 c		0.62 ± 0.01 c		0.71 ± 0.03 c		0.71 ± 0.03 c	
	8	/	0.12 ± 0.08 ab		0.11 ± 0.06 ab		0.47 ± 0.01 b		0.47 ± 0.01 b		0.47 ± 0.02 b		0.47 ± 0.03 b	
	10	/	0.06 ± 0.01 a		0.06 ± 0 a		0.31 ± 0.03 a		0.32 ± 0.03 a		0.40 ± 0.01 a		0.39 ± 0.01 a	
D	15	0.2%	0.03 ± 0 b	<0.001	0.03 ± 0 b	<0.001	0.42 ± 0.02 b	<0.001	0.39 ± 0.02 b	<0.001	0.41 ± 0.03 b	<0.001	0.39 ± 0.03 b	<0.001
	15	0.5%	0.02 ± 0 a		0.02 ± 0 a		0.19 ± 0.01 b		0.19 ± 0.01 b		0.06 ± 0.01 b		0.06 ± 0.01 b	
	15	1%	0.02 ± 0 a		0.02 ± 0 a		0.16 ± 0 a		0.15 ± 0 a		0.03 ± 0.01 a		0.03 ± 0.01 a	
E	15	0.2%	0.03 ± 0 b	<0.001	0.03 ± 0 b	<0.001	0.30 ± 0.01 c	<0.001	0.29 ± 0 c	<0.001	0.02 ± 0 b	<0.001	0.02 ± 0 b	<0.001
	15	0.5%	0.02 ± 0 b		0.02 ± 0 b		0.27 ± 0.01 b		0.26 ± 0.01 b		0.02 ± 0 b		0.02 ± 0 a	
	15	1%	0.02 ± 0 a		0.02 ± 0 a		0.21 ± 0 a		0.20 ± 0 a		0.02 ± 0 a		0.01 ± 0 a	
F	15	0.2%	0.05 ± 0 c	<0.001	0.05 ± 0 b	<0.001	0.18 ± 0 b	<0.001	0.18 ± 0 c	<0.001	0.03 ± 0 b	<0.001	0.03 ± 0 c	<0.001
	15	0.5%	0.05 ± 0 b		0.05 ± 0 a		0.18 ± 0 a		0.16 ± 0 b		0.02 ± 0 a		0.02 ± 0 b	
	15	1%	0.03 ± 0 a		0.03 ± 0 a		0.01 ± 0 a		0.01 ± 0 a		0.01 ± 0 a		0.01 ± 0 a	
G	15	0.2%	0.31 ± 0.01 b	<0.001	0.31 ± 0.01 b	<0.001	0.75 ± 0.03 c	<0.001	0.74 ± 0.03 c	<0.001	0.76 ± 0.01 b	0.001	0.77 ± 0.02 b	0.001
	15	0.5%	0.28 ± 0.01 a		0.27 ± 0.02 a		0.65 ± 0 b		0.63 ± 0.02 b		0.66 ± 0.06 a		0.67 ± 0.07 a	
	15	1%	0.09 ± 0.02 a		0.09 ± 0.02 a		0.59 ± 0.02 a		0.57 ± 0.01 a		0.47 ± 0.04 a		0.46 ± 0.05 a	

A: Water soaking and peeling. B: Heated water soaking and peeling. C: Ultrasound-assisted soaking and peeling. D: Water soaking with NaCl and peeling. E: Water soaking with AA and peeling. F: Water soaking with LAS and peeling. G: Water soaking with EA and peeling. *p* represented confidence within the 99% confidence level of concentrations of penthiopyrad enantiomers, different lowercase letters indicate statistical significance between different treatments for cucumber by Duncan’s multiple range test (*p* = 0.01).

**Table 4 foods-12-00892-t004:** PF values of racemic penthiopyrad in cucumber samples after different soaking methods.

Treatment	Duration (min)	Additive Content	PF (Average ± SD, *n* = 3)					
			Cucumber					
			Pulp	*p*	Peel	*p*	Whole Fruit	*p*
A	5	/	0.06 ± 0 c	<0.001	0.89 ± 0.03 d	<0.001	0.89 ± 0.02 d	<0.001
	10	/	0.05 ± 0 c		0.68 ± 0.06 c		0.79 ± 0.01 c	
	15	/	0.04 ± 0 b		0.40 ± 0.03 b		0.39 ± 0.03 b	
	20	/	0.04 ± 0 a		0.11 ± 0 a		0.20 ± 0.02 a	
B	5	/	0.17 ± 0.01 d	<0.001	0.31 ± 0 c	<0.001	0.06 ± 0.01 c	<0.001
	10	/	0.11 ± 0 c		0.28 ± 0.01 b		0.04 ± 0 bc	
	15	/	0.09 ± 0.01 b		0.22 ± 0.04 a		0.03 ± 0 b	
	20	/	0.03 ± 0 a		0.15 ± 0.01 a		0.02 ± 0 a	
C	4	/	0.43 ± 0.21 b	0.001	0.81 ± 0.04 d	<0.001	0.84 ± 0.03 d	<0.001
	6	/	0.35 ± 0.04 ab		0.61 ± 0.01 c		0.71 ± 0.02 c	
	8	/	0.12 ± 0.07 ab		0.47 ± 0.01 b		0.47 ± 0.02 b	
	10	/	0.06 ± 0 a		0.31 ± 0.03 a		0.39 ± 0.01 a	
D	15	0.2%	0.03 ± 0 b	<0.001	0.40 ± 0.02 b	<0.001	0.40 ± 0.03 b	<0.001
	15	0.5%	0.02 ± 0 a		0.19 ± 0.0 ab		0.06 ± 0.01 b	
	15	1%	0.02 ± 0 a		0.15 ± 0 a		0.03 ± 0.01 a	
E	15	0.2%	0.03 ± 0 b	<0.001	0.30 ± 0 c	<0.001	0.02 ± 0 c	<0.001
	15	0.5%	0.02 ± 0 b		0.27 ± 0.01 b		0.02 ± 0 b	
	15	1%	0.02 ± 0 a		0.21 ± 0 a		0.01 ± 0 a	
F	15	0.2%	0.05 ± 0 c	<0.001	0.18 ± 0 c	<0.001	0.03 ± 0 c	<0.001
	15	0.5%	0.05 ± 0 b		0.17 ± 0 b		0.02 ± 0 b	
	15	1%	0.03 ± 0 a		0.01 ± 0 a		0.01 ± 0 a	
G	15	0.2%	0.31 ± 0.01 b	<0.001	0.74 ± 0.03 c	<0.001	0.77 ± 0.02 b	<0.001
	15	0.5%	0.28 ± 0.01 a		0.64 ± 0.01 b		0.67 ± 0.07 a	
	15	1%	0.09 ± 0.02 a		0.58 ± 0.01 a		0.47 ± 0.04 a	

A: Water soaking and peeling. B: Heated water soaking and peeling. C: Ultrasound-assisted soaking and peeling. D: Water soaking with NaCl and peeling. E: Water soaking with AA and peeling. F: Water soaking with LAS and peeling. G: Water soaking with EA and peeling. *p* represented confidence within the 99% confidence level of concentrations of penthiopyrad enantiomers, different lowercase letters indicate statistical significance between different treatments for tomato by Duncan’s multiple range test (*p* = 0.01).

## Data Availability

Data is contained within the article and Appendix A.

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
