# Peer review of "Reduction in the Residues of Penthiopyrad in Processed Edible Vegetables by Various Soaking Treatments and Health Hazard Evaluation in China"

_foods, 2023, doi:10.3390/foods12040892_

Round 1

Reviewer 1 Report

The manuscript titled “Reduction in the residues of penthiopyrad in processed edible vegetables by various soaking treatments and health hazard evaluation in China” By Chang et al., submitted to “Foods” journal aims to evaluate the removal efficiency of penthiopyrad residues from tomato and cucumber samples by regular and ultrasound-assisted soaking. The removal of penthiopyrad was also investigated by regular soaking in different solutions (heated aqueous, inorganic salt, acid, surfactant, and organic solutions). Meanwhile, among the above trials, the distribution and health hazard risk of penthiopyrad in different parts of two edible vegetables (pulp, peel and whole fruit) and the persistence in tomato and cucumber productions (sauce or juice) were illustrated.

Some comments are raised in the PDF file of the manuscript. Please go through consider them in your revision.

Author Response

The manuscript titled “Reduction in the residues of penthiopyrad in processed edible vegetables by various soaking treatments and health hazard evaluation in China” By Chang et al., submitted to “Foods” journal aims to evaluate the removal efficiency of penthiopyrad residues from tomato and cucumber samples by regular and ultrasound-assisted soaking. The removal of penthiopyrad was also investigated by regular soaking in different solutions (heated aqueous, inorganic salt, acid, surfactant, and organic solutions). Meanwhile, among the above trials, the distribution and health hazard risk of penthiopyrad in different parts of two edible vegetables (pulp, peel and whole fruit) and the persistence in tomato and cucumber productions (sauce or juice) were illustrated.

Some comments are raised in the PDF file of the manuscript. Please go through consider them in your revision.

  1. Please start a new paragraph from here.

Answer: Thank you for your suggestion. We have started from the recommended sentence and the revision was shown in the revised paper.

  1. A point with details for the preparation of samples for LC-MS/MS analysis is needed under M&M section.

Answer: Thank you for your suggestion. We have separated the “Instrument and pretreatment” section to “Instrument” and “Pretreatment” sections. The details were listed in each section.

On line 82-88 of page 2 in the previous paper, “2.2. Instrumentation and pretreatment

Raw and processed tomato and cucumber samples were thoroughly crushed and homogenized for subsequent analysis. Chromatographic separation and quantification of penthiopyrad enantiomers in different matrices were performed on a liquid chromatography with tandem mass spectrometry (LC-MS/MS) and the pretreatment method was a modified QuEChERS (quick, easy, cheap, effective, rugged and safe) approach which has been reported in our previous research [11].” was changed to “2.2. Instrumentation

Chromatographic separation and quantification of penthiopyrad enantiomers in different matrices were performed on a liquid chromatography with tandem mass spectrometry (LC-MS/MS) [11]. The conditions of LC system (Shimadzu Corporation, Kyoto, Japan) included: column, Superchiral S-OD chiral column (150×4.6 mm i.d.; particle size, 5 µm; Shanghai Chiralway Biotech Co., Ltd., Shanghai, China); mobile phase, 0.1% FA in ACN/0.1% FA in aqueous solution (50/50, v/v); flow rate, 1.0 mL/min; injection volume, 3 μL; column temperature, 30 °C; retention time, 11 min. The conditions of MS system (Applied Biosystems, Foster City, USA) were: gas source, nitrogen (99.999%); ion source temperature, 600 °C; ion source gas 1 and 2 pressure, 448 kPa; ion spray voltage, 5.5 kV; detection mode, positive electron spray ionization and multiple reaction monitoring; m/z (parent ion→daughter ion), 360.3→256.0 for confirmation and 360.3→276.0 for quantification; declustering potential, 103.5 V; entrance potential, 10.0 V; collision energy and collision cell exit potential, 28.7 V and 15.7 V for daughter ion 256.0 and 18.4 V and 25.9 V for daughter ion 276.0, respectively.

2.3. Pretreatment

Raw and processed tomato and cucumber samples were thoroughly crushed and homogenized for subsequent analysis. The pretreatment method was a modified QuEChERS (quick, easy, cheap, effective, rugged and safe) approach which has been reported in our previous research [11]. Ten gram of tomato (or cucumber) sample was weighed in a centrifuge tube and then 20 mL of ACN, 2 g of NaCl and 4 g of anhydrous Na2SO4 were added. After vortex-mixing for 5 min and centrifuging for 5 min at 1677 × g, 1 mL of supernatant was transferred to a centrifuge tube containing 50 mg of PSA. The mixture was vortexed for 0.5 min, centrifuged for 3 min at 6708 × g and filtered through a nylon filter syringe before instrument analysis.”.

  1. what is the concentration?

Answer: Thank you for your suggestion. The concentration was 0.5 mg/L and we have added the concentration in the revised paper. On line 94 of page 3 in the previous paper, “5 L of penthiopyrad solution” was changed to “5 L of penthiopyrad solution (0.5mg/L)”.

  1. for how long?

Answer: Thank you for your suggestion. The dipping duration was 60 min and we have added the duration in the revised paper. On line 94-95 of page 3 in the previous paper, “allowing the raw samples for to be fully immersed” was changed to “allowing the raw samples for to be fully immersed for 60 min”.

  1. delete

Answer: Thank you for your suggestion. On line 123 of page 3 in the previous paper, the first “%” was deleted.

  1. delete

Answer: Thank you for your suggestion. On line of page 3 in the previous paper, “was” was deleted.

  1. please indicate what are R- , S- , and Rac- stands for?

Answer: Thank you for your suggestion. We have revised this part and the changes were listed as follows:

On line 138 of page 3 in the previous paper, “R-(−)-enantiomer to the level of Rac-penthiopyrad” was changed to “R-(−)-enantiomer (penthiopyrad enantiomer with a R spatial configuration) to the level of racemic penthiopyrad (Rac-penthiopyrad)”.

  1. please make sure that specified figure is correct.

Answer: Thank you for your suggestion. We have revised the improper description and the figure. On line 200 of page 6 in the previous paper, “saucing in different durations (H)” was changed to “storing process at 4 °C and room temperature for different durations after saucing”. The figure was replaced with a revised one.

  1. please make sure that the specified figure is correct.

Answer: Thank you for your suggestion. We have revised the improper description. On line 200 of page 6 in the previous paper, “juicing in different durations (H)” was changed to “storing process at room temperature 4 °C and for different durations after juicing”. The figure was replaced with a revised one.

Reviewer 2 Report

Manuscript title “Reduction in the residues of penthiopyrad in processed edible vegetables by various soaking treatments and health hazard evaluation in China”

Authors: Chang, Jinming; Dou, Li; Ye, Yu; Zhang, Kankan.

1.     Line 73, section 2.1. Please add the purity and concentrations of all regents used.

2.     Line 82, section 2.2. "Instrumentation and pretreatment". It would be better to separate this section. For example, make it 2.2. "Instrumentation" and 2.3. "Pretreatment". Or "Pretreatment" should be added to section 2.3.  The "Instrumentation" section should include detailed information about the instrument. For example: full instrument name and country of manufacture, which reagents and additional gases are used for the instrument (do not forget to write about reagent purity), column name and characteristics, detailed description of analysis conditions (temperature, pressure, etc.), name of the detector used.

3.     Part “2.4. Data analysis”. Which statistical analysis software was used to evaluate all the results?

4.     Part 3.2., Line 231. You showed the data after 12 days of storage. Did you take samples before 12 days? Why did you choose exactly 12 days?

5.     Line 316-320.  Please cite the literature from which this information was taken.

6.     You identified racemic pentiopyrad in two vegetables (in a cucumber and a tomato). What analytical signals did you use to identify the pentiopyrad (chromatogram, spectra, etc.)?

7.     What approach to penthiopyrad calibration was used? Matrix-specific calibration, external calibration, addition of standards?

8.     ‘‘Discussion’’ part: this section should be expanded.

Author Response

Manuscript title “Reduction in the residues of penthiopyrad in processed edible vegetables by various soaking treatments and health hazard evaluation in China”

Authors: Chang, Jinming; Dou, Li; Ye, Yu; Zhang, Kankan.

  1. Line 73, section 2.1. Please add the purity and concentrations of all regents used.

Answer: Thank you for your suggestion. We have added the purity or concentration of all regents and the changes were listed as follows:

On line 75-83 of page 2 in the previous paper, “Racemic penthiopyrad (purity, 99.5%), manufactured by Dr. Ehrenstorfer GmbH (Augsburg, Germany), was purchased from J&K Scientific Ltd. (Beijing, China). Chromatographic grade acetonitrile (ACN), methanol (MeOH) and formic acid (FA) were obtained from Thermo Fisher Scientific (Waltham, USA). Analytical grade sodium chloride (NaCl), ethanol (EA), linear alklybezene sulfonates (LAS) and acetic acid (AA) were provided by Tianjin Zhiyuan Reagent Co., Ltd. (Tianjin, China). PSA (primary secondary amine) was bought from Agela Technologies (Tianjin, China) and 0.22 μm nylon syringe filter was obtained from PeakSharp Technologies (Beijing, China).” was changed to “Racemic penthiopyrad (purity, 99.5%), manufactured by Dr. Ehrenstorfer GmbH (Augsburg, Germany), was purchased from J&K Scientific Ltd. (Beijing, China). Chro-matographic grade acetonitrile (ACN, 99.9%), methanol (MeOH, 99.9%) and formic acid (FA, 8.0%) were obtained from Thermo Fisher Scientific (Waltham, USA). Analytical grade sodium chloride (NaCl, 99.5%), anhydrous sodium sulfate (Na2SO4, 99.0%), ethanol (EA, 99.7%), linear alklybezene sulfonates (LAS, 99.0%) and acetic acid (AA, 99.5%) were provided by Tianjin Zhiyuan Reagent Co., Ltd. (Tianjin, China). Primary secondary amine (PSA, 99.5%) was bought from Agela Technologies (Tianjin, China) and 0.22 μm nylon syringe filter was obtained from PeakSharp Technologies (Beijing, China).”.

  1. Line 82, section 2.2. "Instrumentation and pretreatment". It would be better to separate this section. For example, make it 2.2. "Instrumentation" and 2.3. "Pretreatment". Or "Pretreatment" should be added to section 2.3.  The "Instrumentation" section should include detailed information about the instrument. For example: full instrument name and country of manufacture, which reagents and additional gases are used for the instrument (do not forget to write about reagent purity), column name and characteristics, detailed description of analysis conditions (temperature, pressure, etc.), name of the detector used.

Answer: Thank you for your suggestion. We have separated the “Instrument and pretreatment” section to “Instrument” and “Pretreatment” sections. The details were listed in each section.

On line 82-88 of page 2 in the previous paper, “2.2. Instrumentation and pretreatment

Raw and processed tomato and cucumber samples were thoroughly crushed and homogenized for subsequent analysis. Chromatographic separation and quantification of penthiopyrad enantiomers in different matrices were performed on a liquid chromatography with tandem mass spectrometry (LC-MS/MS) and the pretreatment method was a modified QuEChERS (quick, easy, cheap, effective, rugged and safe) approach which has been reported in our previous research [11].” was changed to “2.2. Instrumentation

Chromatographic separation and quantification of penthiopyrad enantiomers in different matrices were performed on a liquid chromatography with tandem mass spectrometry (LC-MS/MS) [11]. The conditions of LC system (Shimadzu Corporation, Kyoto, Japan) included: column, Superchiral S-OD chiral column (150×4.6 mm i.d.; particle size, 5 µm; Shanghai Chiralway Biotech Co., Ltd., Shanghai, China); mobile phase, 0.1% FA in ACN/0.1% FA in aqueous solution (50/50, v/v); flow rate, 1.0 mL/min; injection volume, 3 μL; column temperature, 30 °C; retention time, 11 min. The conditions of MS system (Applied Biosystems, Foster City, USA) were: gas source, nitrogen (99.999%); ion source temperature, 600 °C; ion source gas 1 and 2 pressure, 448 kPa; ion spray voltage, 5.5 kV; detection mode, positive electron spray ionization and multiple reaction monitoring; m/z (parent ion→daughter ion), 360.3→256.0 for confirmation and 360.3→276.0 for quantification; declustering potential, 103.5 V; entrance potential, 10.0 V; collision energy and collision cell exit potential, 28.7 V and 15.7 V for daughter ion 256.0 and 18.4 V and 25.9 V for daughter ion 276.0, respectively.

2.3. Pretreatment

Raw and processed tomato and cucumber samples were thoroughly crushed and homogenized for subsequent analysis. The pretreatment method was a modified QuEChERS (quick, easy, cheap, effective, rugged and safe) approach which has been reported in our previous research [11]. Ten gram of tomato (or cucumber) sample was weighed in a centrifuge tube and then 20 mL of ACN, 2 g of NaCl and 4 g of anhydrous Na2SO4 were added. After vortex-mixing for 5 min and centrifuging for 5 min at 1677 × g, 1 mL of supernatant was transferred to a centrifuge tube containing 50 mg of PSA. The mixture was vortexed for 0.5 min, centrifuged for 3 min at 6708 × g and filtered through a nylon filter syringe before instrument analysis.”.

  1. Part “2.4. Data analysis”. Which statistical analysis software was used to evaluate all the results?

Answer: Thank you for your suggestion. The statistical analysis software used in this paper was SPSS version 18.0 (SPSS Inc. Chicago, IL, USA), which was shown in the “Data analysis” section.

  1. Part 3.2., Line 231. You showed the data after 12 days of storage. Did you take samples before 12 days? Why did you choose exactly 12 days?

Answer: Thank you for your suggestion. According to some previous literatures (listed in the following “References”), the physical and chemical properties of tomato paste and cucumber juice during storage have been evaluated. During the storage of cucumber juice for an interval of zero, two, four, and six months, the phenolic compounds and flavonoids were significantly (P ≤ 0.05) decreased. Consequently, the potential activity of the juice was reduced; in addition, pH and vitamin C were significantly (P ≤ 0.05) decreased during the storage period. Meanwhile, the TSS and Titratable acidity were significantly raised. According to another document, after 12 months of storage, the total soluble solids of tomato paste significantly increased to and the acidity decreased. During storage at room temperature, the total sugar significantly decreased and the reducing sugar significantly increased. Vitamin C content also decreased significantly during storage. All these physical and chemical changes are small under the condition of cold storage temperature. Pure cucumber juice and tomato juice are vulnerable to microbial attack due to their high moisture content, and microorganisms can also survive in acidic juice at normal temperature or cold storage conditions, so the storage time is not easy to be too long. According to the reported kinetic models, the changes of the physical (weight loss, color and hardness), chemical (titratable acidity, total soluble solids, sugar to acid ratio and pH) and nutritional (total lycopene and carotenoid) quality parameters of cucumber and tomato stored for 12 days were relative stable. Therefore, the storage durations of processed cucumber juice and tomato sauce were 0-12 days.

References:

El-Saadony, M.; Elsadek, M.F.; Mohamed, A.S.; Taha, A.E.; Ahmed, B.M.; Saad, A.M. Effects of chemical and natural additives on cucumber juice’s quality, shelf life, and safety. Foods. 2020, 9, 639.

Kumar, K.; Ray, A.B. Development and shelf-life evaluation of tomato-mushroom mixed ketchup. J. Food Sci. Technol. 2016, 53, 2236–2243.

Al-Dairi, M.; Pathare, P.B.; Al-Yahyai, R. Quality changes kinetic of tomato during transportation and storage. J. Food Process Eng. 2021, 44, e13808.

Take a sample (0 day) two hours after making tomato sauce and cucumber juice, store at 4℃ and room temperature respectively, and take samples at 1, 2, 3, 5, 7, 9 and 12 days.

  1. Line 316-320.  Please cite the literature from which this information was taken.

Answer: Thank you for your suggestion. We have added the references contained the information shown on line 316-320 and made some changes.

On line 316-320 of page in the previous paper, “Tomato sauce contains many acetic compounds, minerals and pectin, which can remove pesticides in several ways. Unlike cucumber juice, which contains only vitamins and dietary fiber, there is little relief from pesticides. The PF data also indicates that storage may remove a certain amount of penthiopyrad, and the effect is more significant in the tomato sauce sample.” was changed to “Tomato sauce contains many acetic compounds, minerals and pectin, which can remove pesticides in several ways [38]. Unlike cucumber juice, which contains only vitamins and dietary fiber, there is little relief from pesticides. The PF data also indicates that storage may remove a certain amount of penthiopyrad, and the effect is more significant in the tomato sauce sample [39].”.

In the “References” section, “38. Prakash, A.; Prabhudev, S.H.; Vijayalaskshmi, M.R.; Prakash, M.; Baskaran, R. Implication of processing and differential blending on quality characteristics in nutritionally enriched ketchup (Nutri-Ketchup) from acerola and tomato. J. Food Sci. Technol. 2015, 53, 3175–3185. https://doi.org/10.1007/s13197-016-2291-z.” and “39. Naman, M.; Masoodi, F.A.; Wani, S.M.; Ahad, T. Changes in concentration of pesticide residues in fruits and vegetables during household processing. Toxicol. Rep. 2022, 9, 1419–1425. https://doi.org/10.1016/j.toxrep.2022.06.013.” were added.

  1. You identified racemic penthiopyrad in two vegetables (in a cucumber and a tomato). What analytical signals did you use to identify the penthiopyrad (chromatogram, spectra, etc.)?

Answer: Thank you for your suggestion. We have used LC-MS/MS to determine penthiopyrad enantiomers in solvent and matrices. LC was used to separate the analyte from the interference, and MS was applied to confirm and quantify the presentation of penthiopyrad. Therefore, spectra were used to identify penthiopyrad.

  1. What approach to penthiopyrad calibration was used? Matrix-specific calibration, external calibration, addition of standards?

Answer: Thank you for your suggestion. Matrix-specific calibration was used to determine the concentration of penthiopyrad, which could reduce the influence of matrix effect caused by the MS instrument and matrices themselves.

  1. ‘‘Discussion’’ part: this section should be expanded.

Answer: Thank you for your suggestion. We have expanded the “Discussion” section and the changes were listed as follows:

On line 271-331 of page 13-14 in the previous paper, the “Discussion” section was changed to “In the current study, the removal efficiency of different soaking (with or without peeling) methods was investigated for penthiopyrad from two edible vegetables. The reduction data illustrates that the soaking duration can improve the removal efficiency of the soaking method for penthiopyrad from tomato samples. Various soaking approaches have been proposed for the target compound with various removal efficiencies. Water (unheated) soaking could remove a part of penthiopyrad residues from tomato and cucumbers and the reduction amount increased in heated water soaking treatment, which demonstrated that temperature could affect the removal of soaking for penthiopyrad. Temperature and ultrasound were positively correlated to the removal efficiency of soaking for penthiopyrad in tomato. Kaushik et al. [31] reported that washing of tomato samples in warm water could increase the relief of tetranilipole by approximately 1.3 times. Heshmati and Nazemi [32] studied the effect of ultrasound-assisted washing to remove the dichlorvos residues from tomato samples and found an increase of removal under ultrasonic conditions. However, the residues of penthiopyrad in tomato pulp samples have not reduced more after treatment B and C, possibly due to the physicochemical properties of penthiopyrad [20]. The exact reasons is unclear and needs to be revealed in our future studies. Adding some chemicals to water could change the removal efficiency of soaking for pesticides from vegetable samples [33]. The residues of penthiopyrad enantiomers decreased after soaking in water with NaCl (treatment D), AA (treatment E), LAS (treatment F) and EA (treatment G), which meant these chemicals could remove penthiopyrad from tomato by soaking. Previous papers illustrated that after adding some additives (including organic acid, chlorine-based and detergent) washing could remove more pesticide residues from tomato [34] and citrus samples [35]. In the present research, we found that water soaking with organic acid (AA), NaCl and surfactant (LAS) showed excellent relief ability for penthiopyrad enantiomers from tomato samples. For EA, the removal efficiency did not improve a lot, which might be related to the polarity of fungicide and chemicals [3]. All the PF values < 1 indicated that the soaking was able to remove penthiopyrad residues from different parts of the tomato samples regardless of the experimental conditions, and a statistical analysis of the data revealed that treatment A-G was able to reduce the concentration of penthiopyrad from the contaminated tomato sample, demonstrating a significant difference in the treatment of certain soaking.

For cucumber, the removal efficiency is significantly improved when the temperature is increased. However, ultrasound has little effect on increasing penthiopyrad reduction and can even inhibit the removal efficiency. The PF values provide further evidence that temperature significantly affects the removal rate of soaking for penthiopyrad from the cucumber samples and the ultrasound-assisted soaking might be an influence, but not an important one. The possible reason is the special physicochemical properties of cucumber. The cucumber surface has a lot of uneven places, which favors the accumulation of pesticides. While ultrasound accelerates the reduction of pesticides, it also accelerates the adsorption and accumulation of pesticides on the uneven surface of cucumbers. When the soaking duration was overly prolonged, the accumulation rate was greater than the reduction rate, and finally decreased the removal efficiency of water soaking method [20,36]. The addition of chemicals reduced the removal of soaking for penthiopyrad from cucumber samples because of the physicochemical properties of penthiopyrad and EA [22,37]. However, the addition of NaCl, AA and LAS was found to improve the removal rate for penthiopyrad from cucumber samples in the waster soaking with additive trials. As the additive content increases, the removal efficiency improves.

Peeling was able to remove most of the fungicide residue from the contaminated tomato and cucumber samples due to the large reduction in the penthiopyrad levels in the pulp samples. After peeling, tomato sauce and cucumber juice are produced from the pulp of the respective edible vegetables. The peeling results indicated that most of the penthiopyrad residues in tomato and cucumber samples were found in the solid part after squeezing [12]. Due to the specific biochemical and microbiological characteristics of two edible vegetable products, the reduction of penthiopyrad were different in tomato sauce and cucumber juice samples [33]. Tomato sauce contains many acetic compounds, minerals and pectin, which can remove pesticides in several ways [38]. Unlike cucumber juice, which contains only vitamins and dietary fiber, there is little relief from pesticides. The PF data also indicates that storage may remove a certain amount of penthiopyrad, and the effect is more significant in the tomato sauce sample [39].

An enantioselective dissipation of penthiopyrad has been illustrated in tomato and cucumber samples cultivated under field and greenhouse conditions in our previous study [11]. During the storage of tomato sauce, the reduction of penthiopyrad was enantioselective (R-enantiomer reduced preferentially) in tomato sauce samples due to EF values > 0.5. The enantioselectivity was perhaps related to the complex microbial community in tomato sauce [28]. Because tomatoes and cucumbers are often washed and then eaten by humans, consumers should be concerned about the health risks of penthiopyrad. For tomato and cucumber pulp samples, the health risk could be negligible for consumers, meaning that after soaking and peeling, the two edible vegetables are safe for human consumption. For whole fruit samples, consumer could eat two edible vegetables that have only been soaked. High HQ values in peel samples indicated that peeling was an effective processing method to remove penthiopyrad residues from tomato and cucumber samples. The results indicate that the above processing methods to remove penthiopyrad residues from vegetables are crucial to protect humans’ health, strengthen the awareness on food hygiene and safety of consumers, and prevent the occurrence of potential food safety cases.”.

Reviewer 3 Report

1. In the abstract, it is necessary to give a brief explanation of penthiopyrad.

2. Line 17-18 "Peeling can remove approximately 90% of penthiopyrad." remove in what? in both samples or? please clarify!

3. Several diseases caused by consumption of penthiopyrad pesticides must be clearly stated in the introduction.

4. An interesting question and the answer should be part of the discussion of this manuscript "How can this be impactful to the community? and what are the views and arguments of the authors on the application of your research method?, in the sense from the laboratory to the community or the field. Please discuss this in discussion.

5. The conclusion is too long, many sentences that should be discussed but instead are placed in the conclusion. I suggest rewriting the conclusion to be stronger and concise.

Author Response

  1. In the abstract, it is necessary to give a brief explanation of penthiopyrad.

Answer: Thank you for your suggestion. We have added an explanation of penthiopyrad in the “Abstract” section. The changes were listed as follows:

On line 10-12 of page 1 in the previous paper, “Tomato and cucumber are two vital edible vegetables that usually appear in people’s daily diet. Different processing methods can remove pesticide residues from vegetables and protect human health.” was changed to “Tomato and cucumber are two vital edible vegetables that usually appear in people’s daily diet. Penthiopyrad is a new type of amide chiral fungicide, which is often used for disease control of vegetables (including tomato and cucumber) due to its wide bactericidal spectrum, low toxicity, good penetration and strong internal absorption. Extensive application of penthiopyrad may have caused potential pollutions in ecosystem. Different processing methods can remove pesticide residues from vegetables and protect human health.”.

  1. Line 17-18 "Peeling can remove approximately 90% of penthiopyrad." remove in what? in both samples or? please clarify!

Answer: Thank you for your suggestion. We have clarified this unclear description. On line 17-18 of page 1 in the previous paper, “Peeling can remove approximately 90% of penthiopyrad.” was changed to “Peeling can remove approximately 90% of penthiopyrad from contaminated tomato and cucumber samples.”.

  1. Several diseases caused by consumption of penthiopyrad pesticides must be clearly stated in the introduction.

Answer: Thank you for your suggestion. We have added the related description and the changes were listed as follows:

On line 38-40 of page 1 in the previous paper, “Penthiopyrad, a chiral succinate dehydrogenase inhibitor (SDHI) fungicide, has broad-spectrum anti-bactericidal activity for tomato, cucumber and other edible vegetables [7-10].” was changed to “Penthiopyrad, a chiral succinate dehydrogenase inhibitor (SDHI) fungicide, has broad-spectrum anti-bactericidal activity for tomato, cucumber and other edible vegetables. It can be effectively control powdery mildew, fruit tree black spot, rust, black star fungus and gray mold [7-10].”.

  1. An interesting question and the answer should be part of the discussion of this manuscript "How can this be impactful to the community? and what are the views and arguments of the authors on the application of your research method?, in the sense from the laboratory to the community or the field. Please discuss this in discussion.

Answer: Thank you for your suggestion. We have added the recommended discussion and further expanded the “Discussion” section and the changes were listed as follows:

On line 271-331 of page 13-14 in the previous paper, the “Discussion” section was changed to “In the current study, the removal efficiency of different soaking (with or without peeling) methods was investigated for penthiopyrad from two edible vegetables. The reduction data illustrates that the soaking duration can improve the removal efficiency of the soaking method for penthiopyrad from tomato samples. Various soaking approaches have been proposed for the target compound with various removal efficiencies. Water (unheated) soaking could remove a part of penthiopyrad residues from tomato and cucumbers and the reduction amount increased in heated water soaking treatment, which demonstrated that temperature could affect the removal of soaking for penthiopyrad. Temperature and ultrasound were positively correlated to the removal efficiency of soaking for penthiopyrad in tomato. Kaushik et al. [31] reported that washing of tomato samples in warm water could increase the relief of tetranilipole by approximately 1.3 times. Heshmati and Nazemi [32] studied the effect of ultrasound-assisted washing to remove the dichlorvos residues from tomato samples and found an increase of removal under ultrasonic conditions. However, the residues of penthiopyrad in tomato pulp samples have not reduced more after treatment B and C, possibly due to the physicochemical properties of penthiopyrad [20]. The exact reasons is unclear and needs to be revealed in our future studies. Adding some chemicals to water could change the removal efficiency of soaking for pesticides from vegetable samples [33]. The residues of penthiopyrad enantiomers decreased after soaking in water with NaCl (treatment D), AA (treatment E), LAS (treatment F) and EA (treatment G), which meant these chemicals could remove penthiopyrad from tomato by soaking. Previous papers illustrated that after adding some additives (including organic acid, chlorine-based and detergent) washing could remove more pesticide residues from tomato [34] and citrus samples [35]. In the present research, we found that water soaking with organic acid (AA), NaCl and surfactant (LAS) showed excellent relief ability for penthiopyrad enantiomers from tomato samples. For EA, the removal efficiency did not improve a lot, which might be related to the polarity of fungicide and chemicals [3]. All the PF values < 1 indicated that the soaking was able to remove penthiopyrad residues from different parts of the tomato samples regardless of the experimental conditions, and a statistical analysis of the data revealed that treatment A-G was able to reduce the concentration of penthiopyrad from the contaminated tomato sample, demonstrating a significant difference in the treatment of certain soaking.

For cucumber, the removal efficiency is significantly improved when the temperature is increased. However, ultrasound has little effect on increasing penthiopyrad reduction and can even inhibit the removal efficiency. The PF values provide further evidence that temperature significantly affects the removal rate of soaking for penthiopyrad from the cucumber samples and the ultrasound-assisted soaking might be an influence, but not an important one. The possible reason is the special physicochemical properties of cucumber. The cucumber surface has a lot of uneven places, which favors the accumulation of pesticides. While ultrasound accelerates the reduction of pesticides, it also accelerates the adsorption and accumulation of pesticides on the uneven surface of cucumbers. When the soaking duration was overly prolonged, the accumulation rate was greater than the reduction rate, and finally decreased the removal efficiency of water soaking method [20,36]. The addition of chemicals reduced the removal of soaking for penthiopyrad from cucumber samples because of the physicochemical properties of penthiopyrad and EA [22,37]. However, the addition of NaCl, AA and LAS was found to improve the removal rate for penthiopyrad from cucumber samples in the waster soaking with additive trials. As the additive content increases, the removal efficiency improves.

Peeling was able to remove most of the fungicide residue from the contaminated tomato and cucumber samples due to the large reduction in the penthiopyrad levels in the pulp samples. After peeling, tomato sauce and cucumber juice are produced from the pulp of the respective edible vegetables. The peeling results indicated that most of the penthiopyrad residues in tomato and cucumber samples were found in the solid part after squeezing [12]. Due to the specific biochemical and microbiological characteristics of two edible vegetable products, the reduction of penthiopyrad were different in tomato sauce and cucumber juice samples [33]. Tomato sauce contains many acetic compounds, minerals and pectin, which can remove pesticides in several ways [38]. Unlike cucumber juice, which contains only vitamins and dietary fiber, there is little relief from pesticides. The PF data also indicates that storage may remove a certain amount of penthiopyrad, and the effect is more significant in the tomato sauce sample [39].

An enantioselective dissipation of penthiopyrad has been illustrated in tomato and cucumber samples cultivated under field and greenhouse conditions in our previous study [11]. During the storage of tomato sauce, the reduction of penthiopyrad was enantioselective (R-enantiomer reduced preferentially) in tomato sauce samples due to EF values > 0.5. The enantioselectivity was perhaps related to the complex microbial community in tomato sauce [28]. Because tomatoes and cucumbers are often washed and then eaten by humans, consumers should be concerned about the health risks of penthiopyrad. For tomato and cucumber pulp samples, the health risk could be negligible for consumers, meaning that after soaking and peeling, the two edible vegetables are safe for human consumption. For whole fruit samples, consumer could eat two edible vegetables that have only been soaked. High HQ values in peel samples indicated that peeling was an effective processing method to remove penthiopyrad residues from tomato and cucumber samples. The results indicate that the above processing methods to remove penthiopyrad residues from vegetables are crucial to protect humans’ health, strengthen the awareness on food hygiene and safety of consumers, and prevent the occurrence of potential food safety cases.”.

  1. The conclusion is too long, many sentences that should be discussed but instead are placed in the conclusion. I suggest rewriting the conclusion to be stronger and concise.

Answer: Thank you for your suggestion. We have rewritten the conclusion part and the changes were listed as follows:

On line 333-358 of page 14 in the previous paper, “In the current study, the removal efficiency of different soaking (with or without peeling) methods was investigated for penthiopyrad from two edible vegetables. Peeling was able to remove most of the fungicide residue from the contaminated tomato and cucumber samples due to the large reduction in the penthiopyrad levels in the pulp samples. Various soaking approaches have been proposed for the target compound with various removal efficiencies. Water (unheated) soaking could remove a part of penthiopyrad residues from tomato and cucumbers and the reduction amount increased in heated water soaking treatment, which demonstrated that temperature could affect the removal of soaking for penthiopyrad. The ultrasound-assisted soaking method showed different efficiency for reducing penthiopyrad concentrations in tomato (improvement) and cucumber (inhibition) samples because of the specific physicochemical characteristics of the two vegetables. The addition of NaCl, AA and LAS was found to improve the removal rate for penthiopyrad from tomato and cucumber samples in the waster soaking with additive trials. However, after the addition of EA, the removal efficiency of the soaking of both edible vegetables decreased. The stability of penthiopyrad in tomato sauce and cucumber juice samples was also explored at 4 °C and room temperature. After 12 days of storage, the levels of penthiopyrad were reduced > 36% in the tomato sauce sample and 13.3% to 22.9% in the cucumber juice sample, which may be related to the biochemical and microbiological properties of both products. The reduction of penthiopyrad after most processing treatments was not enantioselective (EF ≈ 0.5) and a preferential removal of R-penthiopyrad was found during the storage process of tomato sauce. In addition, HQ data show that consumers can eat the soaked tomatoes and cucumbers, and that it is safer to peel both edible vegetables. The results will provide some information and data on the removal of penthiopyrad from other edible vegetables, and help consumers choose the most effective and simple processing method for reducing pesticide residues in tomatoes and cucumbers.” was changed to “In summary, the effects of different processing methods (soaking, peeling, juicing and saucing) on penthiopyrad residues were investigated in two edible vegetables (tomato and cucumber). The results showed that water soaking at room temperature could remove a part of penthiopyrad residues from tomato (RR < 33%) and cucumbers (RR < 80%) and the reduction amount for tomato (RR > 60%) and cucumber (RR > 90%) increased in heated water soaking treatment. The ultrasound-assisted soaking method showed different efficiency for reducing penthiopyrad concentrations in tomato (improvement, RR > 70%) and cucumber (inhibition, RR < 62%) samples. PF values (decreased from 0.39 to 0.01) indicated that the addition of NaCl, AA and LAS improved the removal rate for penthiopyrad from tomato and cucumber samples in the soaking trials. However, after the addition of EA, the removal efficiency of the soaking of both edible vegetables decreased (PF increased from 0.39 to 0.47). After 12 days of storage, the levels of penthiopyrad were reduced > 36% in the tomato sauce sample and 13.3% to 22.9% in the cucumber juice sample. The reduction of penthiopyrad after most processing treatments was not enantioselective (EF ≈ 0.5) and a preferential removal of R-penthiopyrad was found during the storage process of tomato sauce. In addition, HQ data show that consumers can eat the soaked tomatoes and cucumbers, and that it is safer to peel both edible vegetables. The results will provide some information and data on the removal of penthiopyrad from other edible vegetables, and help consumers choose the most effective and simple processing method for reducing pesticide residues in tomatoes and cucumbers.”.

Round 2

Reviewer 1 Report

Dear Authors,
Thank you for submitting the revised version of your manuscript titled “Reduction in the residues of penthiopyrad in processed edible vegetables by various soaking treatments and health hazard evaluation in China” By Chang et al.
There are no more comments.

Reviewer 2 Report

This work is clearly focused on reducing penthiopyrad residues in processed edible vegetables through various soaking methods. It can be seen that extensive work has been done to correct this work. I accept all corrections.

Reviewer 3 Report

Accept